# A Mac-2 Binding Protein Glycosylation Isomer-Based Risk Model Predicts Hepatocellular Carcinoma in HBV-Related Cirrhotic Patients on Antiviral Therapy

**DOI:** 10.3390/cancers14205063

**Published:** 2022-10-16

**Authors:** Chien-Hung Chen, Tsung-Hui Hu, Jing-Houng Wang, Hsueh-Chou Lai, Chao-Hung Hung, Sheng-Nan Lu, Cheng-Yuan Peng

**Affiliations:** 1Division of Hepatogastroenterology, Department of Internal Medicine, Kaohsiung Chang Gung Memorial Hospital and College of Medicine, Chang Gung University, Kaohsiung 833253, Taiwan; 2Center for Digestive Medicine, Department of Internal Medicine, China Medical University Hospital, Taichung 404022, Taiwan; 3School of Chinese Medicine, China Medical University, Taichung 404022, Taiwan; 4School of Medicine, China Medical University, Taichung 404022, Taiwan

**Keywords:** AFP, cirrhosis, hepatocellular carcinoma, M2BPGi, risk model

## Abstract

**Simple Summary:**

Mac-2 binding protein glycosylation isomer (M2BPGi) has not been used in a risk score to predict hepatocellular carcinoma (HCC). We enrolled 1003 cirrhotic patients receiving entecavir or tenofovir monotherapy to construct an HCC risk score. The ASPAM-B score, based on age, sex, platelet count, AFP and M2BPGi at 12 months of treatment, was developed. The ASPAM-B scores accurately classified patients into low (0–3.5), medium (4–7) and high (>7) risk (*p* < 0.001). The values of AUROC for predicting 3-, 5- and 9-year risks of HCC were 0.742, 0.728 and 0.719, respectively. All AUROCs between the ASPAM-B and APA-B, PAGE-B, RWS-HCC and THRI scores at 3–9 years were significantly different. The M2BPGi-based risk model exhibited good discriminant function in predicting HCC in cirrhotic patients who received antiviral treatment.

**Abstract:**

Mac-2 binding protein glycosylation isomer (M2BPGi) has not been used in a risk score to predict hepatocellular carcinoma (HCC). We enrolled 1003 patients with chronic hepatitis B and cirrhosis receiving entecavir or tenofovir therapy for more than12 months to construct an HCC risk score. In the development cohort, Cox regression analysis identified male gender, age, platelet count, AFP and M2BPGi levels at 12 months of treatment as independent risk factors of HCC. We developed the HCC risk prediction model, the ASPAM-B score, based on age, sex, platelet count, AFP and M2BPGi levels at 12 months of treatment, with the total scores ranging from 0 to 11.5. This risk model accurately classified patients into low (0–3.5), medium (4–7), and high (>7) risk in the development and validation groups (*p* < 0.001). The areas under the receiver operating characteristic curve (AUROC) of 3-, 5- and 9-year risks of HCC were 0.742, 0.728 and 0.719, respectively, in the development cohort. All AUROC between the ASPAM-B and APA-B, PAGE-B, RWS-HCC and THRI scores at 3–9 years were significantly different. The M2BPGi-based risk model exhibited good discriminant function in predicting HCC in cirrhotic patients who received long-term antiviral treatment.

## 1. Introduction

Long-term treatment with nucleos(t)ide analogues (NA) could reduce rates of cirrhotic complications, hepatocellular carcinoma (HCC), and total or liver-related mortality [1,2]. Although NA treatment could reduce the rate of HCC, it does not eliminate its development, especially in cirrhotic patients [1,2]. In recent years, some risk prediction models of HCC in patients with chronic hepatitis B (CHB) who received long-term NA treatment have been developed [3,4,5,6,7,8,9]. Notably, a recent study from the United States compared the predictive performance of 10 risk prediction models and demonstrated that APA-B, AASL-HCC, REAL-B and RWS-HCC exhibited an area under the receiver operating characteristic curve (AUROC) of >0.80 for predicting 3-year HCC risk [4,6,7,10,11,12,13,14,15]. However, these risk scores were developed from a mixed population with or without cirrhosis. Further validation of these risk models is warranted to determine their clinical utility in cirrhotic patients.

The *Wisteria floribunda* agglutinin (WFA)-positive Mac-2 binding protein glycosylation isomer (M2BPGi), a secreted glycoprotein from hepatic stellate cells (HSCs) in the serum and extracellular matrix, can induce the expression of Mac-2 protein in Kupffer cells, which in turn activates HSCs and increases alpha-smooth muscle actin expression [16]. In recent years, it has been demonstrated that serum M2BPGi levels correlate with the stage of liver fibrosis in patients with CHB and that serum M2BPGi level is a useful marker of HCC in CHB patients receiving NA therapy and a predictor of recurrence and prognosis in patients with HCC undergoing curative resection [17,18,19,20,21,22].

The aim of this study was to study the predictive role of serum M2BPGi for HCC occurrence and developed a new, M2BPGi-based risk model of HCC in a cohort of CHB patients with cirrhosis receiving entecavir or tenofovir disoproxil fumarate (TDF) treatment.

## 2. Materials and Methods

### 2.1. Patients

This study retrospectively enrolled a cohort of 689 CHB patients with cirrhosis who received entecavir treatment between 2008 and 2018, and 314 CHB patients with cirrhosis who received TDF treatment between 2011 and 2018. The patients were included from China Medical University Hospital (*n* = 337) and Kaohsiung Chang Gung Memorial Hospital (*n* = 666). In Taiwan, the costs of entecavir and TDF have been reimbursed for hepatitis B virus (HBV) treatment by Taiwan’s National Health Plan since 2008 and 2011, respectively. The inclusion criteria were: (1) age >18 years and hepatitis B surface antigen (HBsAg) was positive for more than 6 months before NA therapy; (2) entecavir or TDF monotherapy for at least 12 months before enrollment; (3) all patients fulfilled the diagnosis of cirrhosis either liver histology (*n* = 210) or cirrhosis was suggested by repeated ultrasounds and clinical features, such as gastroesophageal varices, splenomegaly, thrombocytopenia or ascites. The exclusion criteria were: (1) evidence of alcoholic liver disease, autoimmune hepatitis, or coinfection with hepatitis C virus (HCV), hepatitis D virus or human immunodeficiency virus; (2) HCC or liver transplantation at baseline or within the first year of NA therapy.

All enrolled patients were randomly assigned to the models of development or validation group in a 2:1 ratio to construct prediction model of HCC. The clinical parameters at baseline and 12 months of treatment were used to construct the HCC prediction model in the development cohort, and validation cohort were used to examine its predictive performance.

### 2.2. Methods

All patients received NA therapy with a median duration of 72 (12–172) months. During NA therapy, all patients were followed up every 1 to 3 months. Serum HBV DNA and alanine aminotransferase (ALT) levels and were checked at baseline, every 3 to 6 months during NA treatment, and at the time of biochemical breakthrough. All enrolled patients were followed until discontinuation of entecavir or TDF treatment or the last visit. HCC surveillance was implemented using serum alpha-fetoprotein (AFP) and abdominal ultrasonography every 3 months. HCC was diagnosed according to the practice guidance of the American Association for the Study of Liver Diseases [23].

### 2.3. Definitions

Diabetes mellitus (DM) was diagnosed according to the previous guideline [24]. Patients were also considered diabetic according to their medical history or if they had received insulin treatment or oral hypoglycemic agents. Hypertension was diagnosed according to the medical history or having received anti-hypertensive drugs. Cirrhotic events were defined as new developments of hepatic encephalopathy, variceal bleeding, or ascites in patients without hepatic decompensation at the initiation of NA treatment.

### 2.4. Measurement of WFA-Positive M2BPGi

Serum WFA-positive M2BP (M2BPGi) level was measured based on a lectin-antibody sandwich immunoassay using the fully automatic immunoanalyzer, HISCL-2000i (Sysmex, Hyogo, Japan) [25]. The values of M2BPGi were expressed as cut-off index (COI) [25].

### 2.5. Serology

Serum HBV DNA was quantified by the COBAS AmpliPrep/COBAS TaqMan HBV test with a detection limit of 20 IU/mL. Hepatitis B core-related antigen (HBcrAg) levels were quantified using the Chemiluminescent Enzyme Immunoassay (CLEIA) system on a Lumipulse CLEIA analyzer (Fujirebio Inc., Tokyo, Japan) following the manufacturer’s instructions [26]. The automated estimation range was from 3 to 7 log U/mL. HBcrAg levels below 3 log U/mL were taken as 3 log_10_ U/mL for statistical analysis.

### 2.6. Statistical Analysis

The cumulative incidences of HCC, cirrhotic events and liver-related mortality were calculated by Kaplan–Meier method with the log-rank test. The risk factors of HCC occurrence, cirrhotic events and mortality were determined by Cox proportional hazards regression model. Missing data were assumed to be missing at random and were replaced with substituted values by multiple imputation [27]. The HCC risk scoring system and HCC risk was established by Cox proportional hazards regression model and the method has been previously described [6,28]. The HCC risk was estimated with the equation: 1–P_0_ ^exp(Σβage×score–Σβ^_i_^×M^_i_^)^. The model discrimination was assessed with area under AUROC curves. AUROCs were calculated by time-dependent ROC curves for accessing the performance of the risk models for each year and used C-statistic to assess the performance of the risk model. The model calibration was compared by Hosmer–Lemeshow goodness-of-fit test between expected and observed rates of HCC in the development group. The time-dependent ROC, C-statistics, and comparisons of these values between two risk scores were performed using the timeROC package [29,30]. A two-sided *p* value of < 0.05 was considered statistically significant.

## 3. Results

### 3.1. Comparison of Clinical Characteristics of All Patients with or without HCC Development

Table 1 compares the clinical features of patients with or without HCC development. We select the variables associated with HCC for analysis according to previous studies [4,6,7,10,11]. Patients with HCC development were older and higher percentages of them were male and more likely had hepatic decompensation and hypertension than those without HCC development. They also had lower albumin levels and platelet counts and higher M2BPGi and HBcrAg levels than those without HCC development.

### 3.2. HCC Risk Predictors and Prediction Model of the Development Group

In the entire cohort, 183 subjects developed HCC during a median follow-up duration of 72 (12–172) months (6153.73 person years). The cumulative HCC incidences at 3, 5, and 10 years were 9.5%, 14.8%, and 25.8%, respectively.

The clinical features of the development and validation cohorts were presented in Appendix A and were similar between two cohorts. The rates of HCC development were 10% versus 9.1% at 3 years, 15% versus 14.3% at 5 years and 25.8% versus 26.1% at 10 years in the development (*n* = 668) and validation groups (*n* = 335), respectively (*p* = 0.845) (Figure 1).

A multivariate analysis showed that sex and age, platelet count, AFP and M2BPGi levels at 12 months of treatment were the independent predictors associated with HCC in the development group (Table 2). The HCC risk prediction model was constructed, basis on age, sex, platelet count, M2BPGi and AFP levels at 12 months of treatment, to develop the risk score, named as ASPAM-B (Table 3). We converted the regression coefficients of the independent risk factors to compute integer risk scores (Table 3). HCC predictive risk scores after 2–10 years of ETV or TDF therapy were listed in Appendix A. Figure 2 shows the nomograms of 3-, 5-, 7- and 9-year risk for hepatocellular carcinoma for this model.

In this model, the total risk scores ranged from 0 to 11.5. The C-statistic of the model was 0.716 (0.665–0.768). The calibration of the model revealed a good model fit (*p* = 0.5661).

We categorized the ASPAM-B score into three subgroups according to the HCC incidence: ≤3.5, 4–7 and >7, respectively. The cumulative HCC rates at 8 years of treatment in the three subgroups were 9.0%, 23.5% and 57.9%, respectively (*p* < 0.001, Figure 3A).

### 3.3. Validation of the HCC Risk Prediction Model

According to ASPAM-B score, validation cohort was also categorized into low (≤3.5), medium (4–7), and high (>7) risk. The cumulative HCC rates at 5 years of treatment in the three subgroups were 5.6%, 24.8% and 44.6%, respectively (*p* < 0.001, Figure 3B). The C-statistic of this risk model was 0.714 (0.647–0.782).

### 3.4. Comparisons of AUROC and C-Statistic between Different Prediction Models of HCC

In the development cohort, the AUROCs for predicting 3-, 5-, 7- and 9-year risks of HCC were 0.742, 0.728, 0.721 and 0.719, respectively, based on ASPAM-B score. The AUROCs for predicting 3-, 5-, 7- and 9-year risks of HCC for the APA-B score [6], PAGE-B score [5], RWS-HCC score [14], AASL-HCC score [10] and Toronto HCC risk index (THRI) [31] are shown in Table 4. All AUROCs between the ASPAM-B and the APA-B, PAGE-B, RWS-HCC and THRI scores at 3–9 years were significantly different (Appendix A).

In the development cohort, the C-statistics of the models of ASPAM-B, APA-B, PAGE-B, RWS-HCC, AASL-HCC and THRI were 0.716 (95% confidence interval [CI]: 0.665–0.768), 0.659 (0.607–0.712), 0.671 (0.620–0.721), 0.618 (0.562–0.673), 0.651 (0.600–0.702), and 0.664 (0.613–0.715), respectively. The ASPAM-B had higher values of C-statistic than APA-B (*p* = 0.011), PAGE-B (*p* = 0.050), RWS-HCC (*p* = 0.0014), AASL-HCC (*p* = 0.0059) and THRI (*p* = 0.020).

### 3.5. Incidences and Predictors of Cirrhotic Events

Among the 814 patients with compensated cirrhosis at baseline, 44 experienced cirrhotic events during treatment, of which 28, 22, and 5 developed ascites, variceal bleeding, and hepatic encephalopathy, respectively. The cumulative incidences of cirrhotic events at 3, 5, and 10 years were 2.9%, 5%, and 8.6%, respectively. A multivariate analysis revealed that lower albumin levels, lower platelet count and higher M2BPGi levels at 12 months of treatment were independent risk factors for cirrhotic events (Table 5). An M2BPGi level of 1.2 COI at 12 months of treatment was the optimal value for predicting cirrhotic events within 10 years (AUROC: 0.819) by time-dependent ROC curve. The 10-year cumulative incidence of cirrhotic events in patients with the M2BPGi level ≤ 1.2 and >1.2 COI were 3.3% and 18.5%, respectively (*p* < 0.001) (Figure 4A).

### 3.6. Incidences and Predictors of Liver-Related Mortality or Liver Transplantation

In the entire cohort, 62 patients developed liver-related mortality during treatment, including 19 patients who underwent liver transplantation. The cumulative incidences of liver-related mortality or liver transplantation at 3, 5, and 10 years were 1.8%, 5.2%, and 10.6%, respectively. A multivariate Cox regression analysis revealed that lower albumin levels, lower platelet count and higher M2BPGi levels at 12 months of treatment were independent predictors for liver-related mortality or liver transplantation (Table 6). AFP tumor biomarker or AST/ALT metabolism makers at baseline or 12 months of treatment were not independent factors of liver related mortality or liver transplantation.

We used an M2BPGi level of 1.2 COI as the optimal cutoff value. The 10-year cumulative incidences of liver-related mortality or liver transplantation in patients with the M2BPGi level ≤ 1.2 and >1.2 COI were 5.4% and 17.5%, respectively (*p* < 0.001) (Figure 4B).

## 4. Discussion

Our study demonstrated that the cumulative HCC rates at 3, 5, and 10 years were 9.5%, 14.8%, and 25.8%, respectively, among patients with CHB and cirrhosis undergoing entecavir or TDF treatment. In the development group, we constructed the ASPAM-B risk score on the basis of age, sex, platelet, AFP and M2BPGi levels at month 12 of treatment, with the total scores ranging of 0 to 11.5. The ASPAM-B score predicted HCC risk over 2 to 9 years with an overall C-statistic of 0.716, which was significantly higher than those of the APA-B, PAGE-B, RWS-HCC, AASL-HCC and THRI, and stratified patients into three subgroups with distinct HCC risks. Furthermore, the stratified risk scores could be verified accurately in the validation cohort (*p* < 0.001).

To date, no risk models were specifically developed to predict HCC in HBV-related patients with cirrhosis, particularly those who have been receiving long-term NA treatment. It would be desirable if one can develop an HBV-specific risk model which enables us to stratify the risk of HCC among such population. 

To our knowledge, at least 10 risk models have been developed to predict the risk of HCC in CHB patients receiving NA therapy [5,6,7,8,9,10,11,12,13,14,15,31,32]. All of them were developed from a mixed population with or without cirrhosis (19.1–50.2% with cirrhosis). A recent cohort study from the United States demonstrated that three models (APA-B, REAL-B and AASL-HCC) developed from patients receiving NA treatment and one model (RWS-HCC) developed from predominantly treatment-naïve patients exhibited the highest AUROCs (all > 0.80) for predicting 3-year HCC risk [6,10,11,14,15]. Common parameters of these four risk models such as age, cirrhosis or platelet count and AFP are documented risk factors for HCC. A recent Korean study revealed that cirrhosis at baseline, platelet count and AFP at 12 months of NA treatment were the optimal predictive factors for HCC in CHB patients receiving entecavir or TDF treatment [13]. A study from Toronto enrolled all cirrhotic patients with mixed etiology showed that age, sex, etiology and platelets were associated with HCC [31]. In the current study, we found that age, sex, platelet count and AFP level at 12 months of treatment were independent predictors associated with HCC occurrence in the development cohort of cirrhotic patients, similar to what we demonstrated in a prior study cohort (APA-B) which comprised only 36% cirrhotic patients [6]. However, the discriminant performance of APA-B for HCC occurrence was less satisfactory in the cirrhotic cohort compared to the mixed population with or without cirrhosis (C-statistic: 0.659 versus 0.850) [6]. Further effort in identification of novel biomarkers into the APA-B model to improve its performance in cirrhotic patients is warranted.

Serum M2BPGi levels correlate with the liver fibrosis stage in CHB patients and could predict HCC occurrence in CHB patients receiving NA treatment [17,18,19,20,21,22]. Given that platelet count shows an inverse correlation with the hepatic venous pressure gradient. It remains to be elucidated whether M2BPGi can serve as a biomarker of HCC risk independently of platelet in cirrhotic patients receiving NA treatment [33,34]. We demonstrated that both M2BPGi level and platelet count at 12 months of NA treatment, rather than at baseline, were independent predictors of HCC occurrence. Incorporation of sex and M2BPGi into the APA-B risk model generated the ASPAM-B model, which improved the discriminative performance for HCC occurrence and yielded significantly higher AUROC values for predicting 3-, 5-, 7- and 9-year HCC risks than did the APA-B model (all *p* < 0.05). The ASPAM-B model also outperformed the PAGE-B, RWS-HCC, AASL-HCC and THRI models with a significantly higher C-statistic for HCC risk prediction. The current demonstration of its predictive role for HCC occurrence independent of platelet in cirrhotic patients suggests that M2BPGi may act in additional unrecognized pathways related to hepatocarcinogenesis, rather than simply serve as a surrogate marker of liver fibrosis. Moreover, in addition to the liver fibrosis stage, serum M2BPGi levels correlate with the degree of hepatic inflammation in patients with CHB [17]. M2BPGi levels at 12 months of treatment may reliably reflect the actual liver fibrosis stage. A previous study also revealed that M2BPGi levels at 12 months of treatment correlated better with future HCC risk compared to the baseline measurement [20]. Thus, we propose that 12 months after NA treatment may represent an ideal time point for future refinement of the optimal risk model to predict HCC occurrence in patients with CHB receiving long-term NA treatment.

Serum HBcrAg reflects the levels of intrahepatic covalently closed circular DNA transcriptional activity [26]. Baseline or on-treatment HBcrAg levels were reported to predict HCC occurrence in CHB patients receiving NA treatment [35]. In the current study, serum HBcrAg level at baseline or 12 months of NA treatment was not a risk factor of HCC occurrence in the entire cohort. 

In addition to HCC, serum M2BPGi level and platelet count at 12 months of treatment also predicted cirrhotic events, liver-related mortality or liver transplantation in patients receiving NA treatment. This finding could be reconciled by the fact that M2BPGi and platelet may reflect the severity of underlying cirrhosis or hepatocyte dysfunction.

The current study has some limitations to note. First, the diagnosis of cirrhosis was confirmed by histology only in 210 patients of the entire cohort. Patients with early cirrhosis might have been excluded from this study. Second, the present model was developed from Asian patients who acquired HBV genotype B or C infection during neonatal period. External validation of the risk model with patients of different ethnicities or HBV genotypes are required to further verify its discriminant performance for HCC occurrence. Third, because we only measured serum M2BPGi levels at 12 months of NA treatment, we could only address the predictive role of serum M2BPGi level at 12 months of treatment in the subsequent risk of HCC in this study. We will explore the predictive role of serum M2BPGi levels at later time points during NA treatment in the subsequent risk of HCC in future studies. Fourth, despite that ASPAM-B exhibited the highest C-statistic for HCC occurrence in cirrhotic patients among all available risk models, its performance was moderate. Future efforts should be directed toward implementing novel biomarkers to facilitate early prediction and diagnosis of HCC in cirrhotic patients.

## 5. Conclusions

The M2BPGi-based ASPAM-B risk model exhibited good discriminant function in predicting HCC occurrence and stratified HCC risk in cirrhotic patients who received long-term NA treatment. Risk stratification of HCC occurrence in these patients may assist clinicians with individualization of HCC surveillance program in the antiviral therapy era. In addition to HCC, M2BPGi level at 12 months of treatment was a useful marker to predict cirrhotic events and liver-related mortality in cirrhotic patients receiving NA treatment.

## Figures and Tables

**Figure 1 cancers-14-05063-f001:**
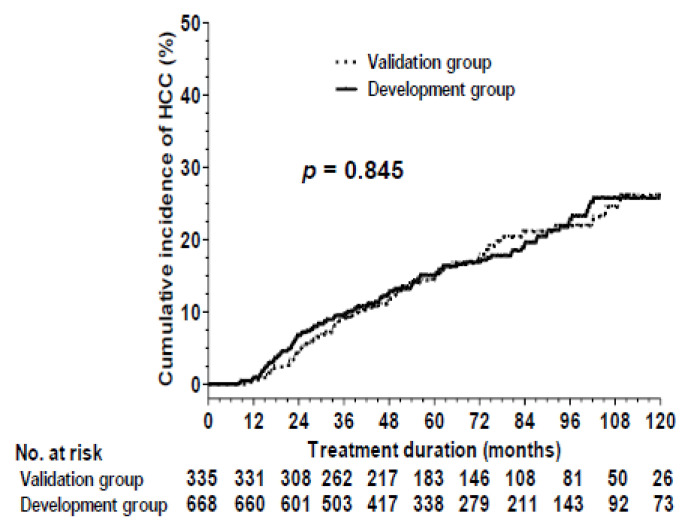
Comparison of cumulative incidences of HCC between development and validation groups.

**Figure 2 cancers-14-05063-f002:**
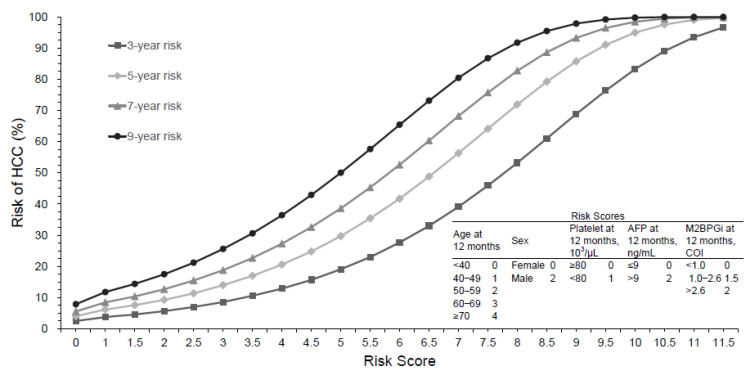
Nomograms for the prediction of the risk of hepatocellular carcinoma development.

**Figure 3 cancers-14-05063-f003:**
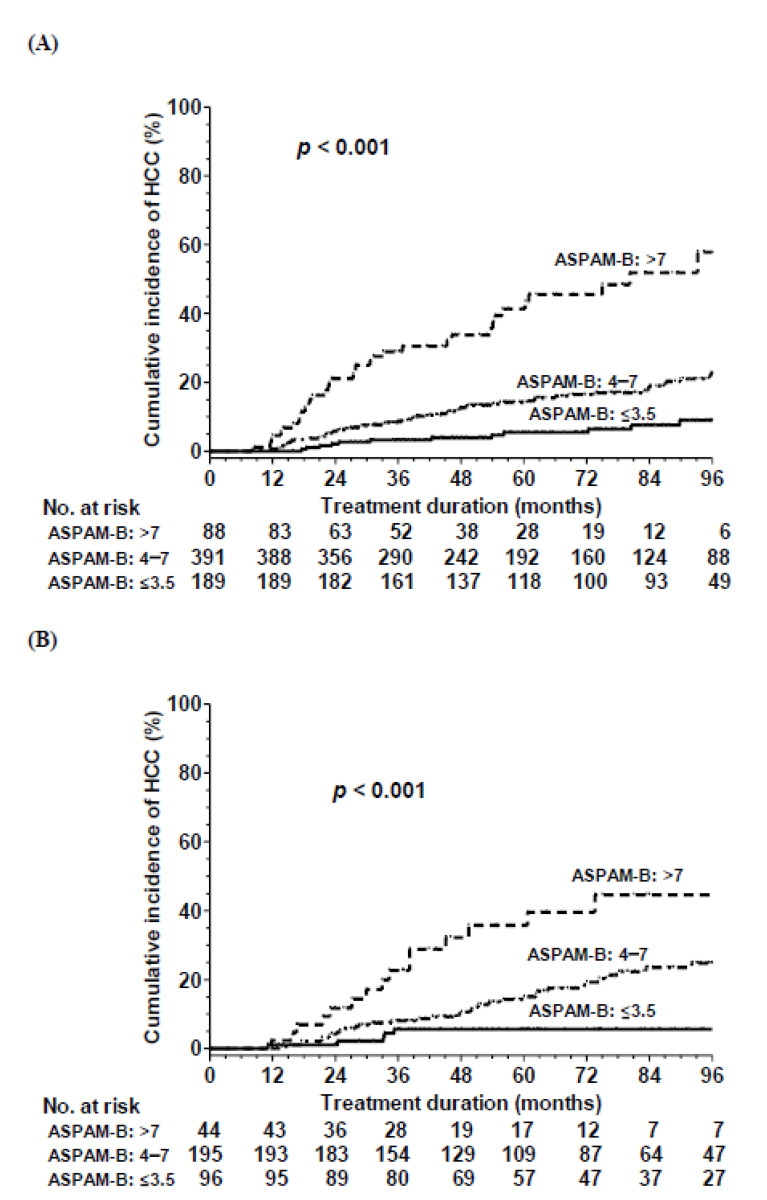
Cumulative incidence rate of HCC according to the ASPAM-B risk score in the (**A**) development and (**B**) validation cohorts.

**Figure 4 cancers-14-05063-f004:**
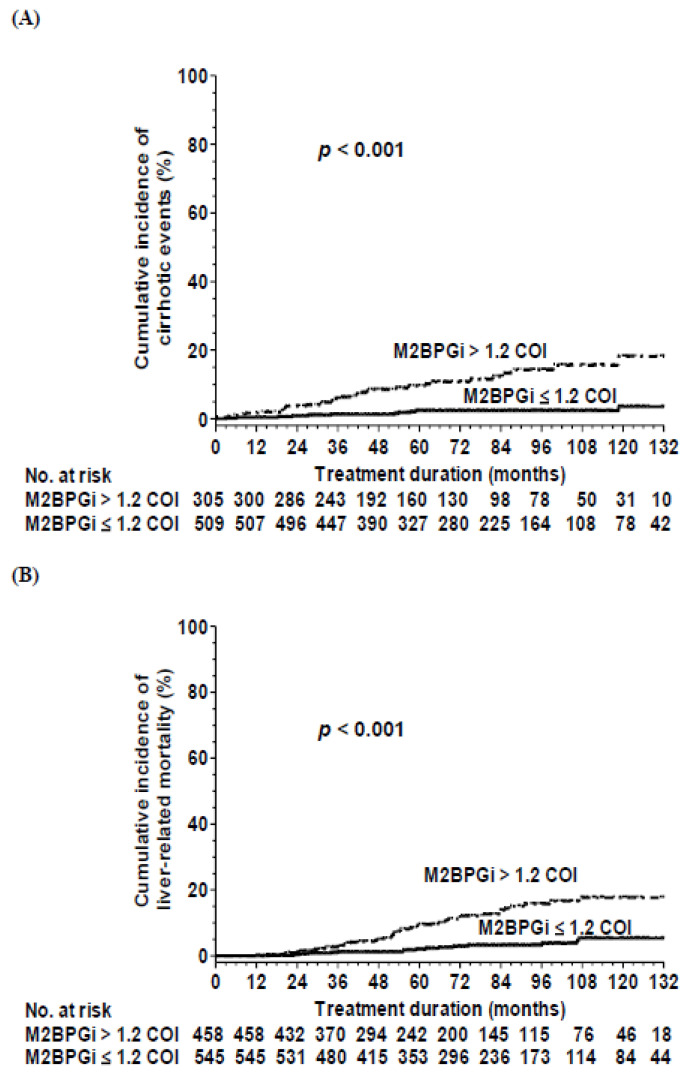
Cumulative incidences of (**A**) cirrhotic events and (**B**) liver-related morality or liver transplantation according to M2BPGi levels at 12 months of treatment.

**Table 1 cancers-14-05063-t001:** Baseline clinical characteristics of patients with HCC or without HCC development.

Variables	HCC *n* = 183	No HCC *n* = 820	*p* Value
Age (year)	58.1 ± 10.0	53.0 ± 12.0	<0.001
Sex, male	147 (80.3%)	599 (73.0%)	0.041
Entecavir versus TDF	147 vs. 36	542 vs. 278	<0.001
HBeAg-positive status	50 (27.3%)	197 (24.0%)	0.349
Decompensation status	49 (26.8%)	140 (17.1%)	0.002
NA-naïve	153 (83.6%)	696 (84.9%)	0.666
Diabetes mellitus, yes	45 (24.6%)	176 (21.5%)	0.356
Hypertension, yes	61 (33.3%)	202 (24.6%)	0.016
HBV DNA, log_10_ IU/mL	5.53 ± 1.52	5.40 ± 1.51	0.318
AST, U/L	119.1 ± 205.4	130.5 ± 269.8	0.589
ALT, U/L	134.2 ± 259.2	162.3 ± 362.8	0.321
Total bilirubin, mg/dL	2.09 ± 3.75	2.03 ± 3.78	0.854
INR	1.19 ± 0.22	1.19 ± 0.28	0.992
Albumin, g/dL	3.80 ± 0.64	4.02 ± 0.63	<0.001
Platelet, ×10^3^/μL	121.0 ± 56.0	138.7 ± 56.0	<0.001
AFP, ng/mL	32.2 ± 82.3	30.2 ± 123.9	0.840
M2BPGi, COI	3.80 ± 3.98	2.77 ± 3.50	<0.001
HBcrAg, log_10_ U/mL	5.33 ± 1.49	5.07 ± 1.48	0.034

Abbreviations: AFP, alpha-fetoprotein; ALT, alanine aminotransferase; AST, aspartate aminotransferase; COI, cut-off index; HBcrAg, hepatitis B core related antigen; HBeAg, hepatitis B e antigen; HCC, hepatocellular carcinoma; INR, international normalized ratio; M2BPGi, Mac-2 binding protein glycosylation isomer; NA, nucleos(t)ide analogue.

**Table 2 cancers-14-05063-t002:** The risk predictors of hepatocellular carcinoma in the development cohort.

	Univariate Analysis		Multivariate Analysis	
Variables	HR (95% CI)	*p* Value	HR (95% CI)	*p* Value
Baseline				
Age (per year)	1.034 (1.0180–1.056)	<0.001		
Sex, male vs. female	1.458 (0.927–2.295)	0.103	2.152 (1.352–3.425)	0.001
HBeAg, yes vs. no	1.051 (0.705–1.566)	0.808		
Decompensation, yes vs. no	1.883 (1.263–2.810)	0.002		
NA-naïve, yes vs. no	0.843 (0.530–1.340)	0.470		
TDF vs. entecavir	0.634 (0.398–1.012)	0.056		
Diabetes mellitus, yes vs. no	1.238 (0.819–1.872)	0.311		
Hypertension, yes vs. no	1.662 (1.151–2.400)	0.007		
HBV DNA, per log_10_ IU/mL	0.959 (0.856–1.075)	0.475		
AST, per U/L	0.999 (0.998–1.000)	0.212		
ALT, per U/L	0.999 (0.999–1.000)	0.120		
Total bilirubin, per mg/dL	0.998 (0.955–1.044)	0.944		
Albumin, per g/L	0.627 (0.484–0.812)	<0.001		
INR, per ratio	0.901 (0.464–1.750)	0.758		
Platelet, per 10^3^/μL	0.993 (0.990–0.997)	<0.001		
AFP, per ng/mL	1.001 (0.999–1.003)	0.375		
M2BPGi, per COI	1.054 (1.013–1.098)	0.010		
HBcrAg, per log_10_ U/mL	1.066 (0.942–1.206)	0.312		
12 months of treatment				
Age (year)	1.034 (1.018–1.050)	<0.001	1.041 (1.024–1.057)	<0.001
ALT < 40 U/L, per U/L	0.685 (0.468–1.001)	0.051		
AFP, per ng/mL	1.009 (1.004–1.015)	<0.001	1.010 (1.005–1.016)	0.003
Platelet, per 10^3^/μL	0.992 (0.988–0.995)	<0.001	0.955 (0.991–0.999)	0.019
M2BPGi, per COI	1.123 (1.069–1.180)	<0.001	1.099 (1.037–1.165)	0.002
HBcrAg, per log_10_ U/mL	1.085 (0.941–1.252)	0.262		

AFP, alpha-fetoprotein; ALT, alanine aminotransferase; AST, aspartate aminotransferase; CI, confidence interval; COI, cut-off index; HBcrAg, hepatitis B core related antigen; HBeAg, hepatitis B e antigen; HR, hazard ratio; INR, international normalized ratio; M2BPGi, Mac-2 binding protein glycosylation isomer; NA, nucleos(t)ide analogue; TDF, Tenofovir disoproxil fumarate.

**Table 3 cancers-14-05063-t003:** The risk scores of hepatocellular carcinoma in the development cohort.

Variables	HR (95% CI)	Parameter	*p* Value	Risk Scores
Age at 12 months, years <40 40–49 50–59 60–69 ≥70	1.532 (1.297–1.809)	0.4265	<0.0001	0 1 2 3 4
Sex Female Male	1.000 2.164 (1.356–3.452)	0.7718	0.0012	0 2
Platelet at 12 months, 10^3^/μL ≥80 <80	1.000 1.779 (1.170–2.706)	0.5760	0.0071	0 1.5
AFP at 12 months, ng/mL ≤9 >9	1.000 2.264 (1.406–3.645)	0.8170	0.0008	0 2
M2BPGi at 12 months, COI <1.0 1.0–2.6 >2.6	1.000 1.904 (1.222–2.967) 2.163 (1.336–3.500)	0.6441 0.7714	0.0044 0.0017	0 1.5 2

AFP, alpha-fetoprotein; CI, confidence interval; COI, cut-off index; HR: hazard ratio; M2BPGi, Mac-2 binding protein glycosylation isomer.

**Table 4 cancers-14-05063-t004:** The values of AUROCs for predicting hepatocellular carcinoma according to different risk models.

	ASPAM-B	APA-B	PAGE-B	RWS-HCC	AASL-HCC	THRI
Development Cohort (*n* = 668)	AUROC (95% CI)	AUROC (95% CI)	AUROC (95% CI)	AUROC (95% CI)	AUROC (95% CI)	AUROC (95% CI)
3 years	0.742 (0.672–0.811)	0.661 (0.588–0.734)	0.673 (0.601–0.746)	0.601 (0.525–0.677)	0.677 (0.611–0.744)	0.660 (0.590–0.731)
5 years	0.728 (0.668–0.788)	0.669 (0.610–0.729)	0.676 (0.616–0.736)	0.604 (0.540–0.668)	0.654 (0.594–0.714)	0.663 (0.604–0.722)
7 years	0.721 (0.665–0.777)	0.668 (0.612–0.724)	0.667 (0.611–0.723)	0.606 (0.546–0.665)	0.644 (0.588–0.701)	0.650 (0.593–0.706)
9 years	0.719 (0.666–0.772)	0.667 (0.614–0.721)	0.671 (0.617–0.724)	0.614 (0.556–0.673)	0.651 (0.598–0.704)	0.656 (0.603–0.710)

AUROC, the area under the receiver operating characteristic curve; CI, confidence interval.

**Table 5 cancers-14-05063-t005:** Univariate and multivariate analyses of factors associated with hepatic events (new events of variceal bleeding, ascites and hepatic encephalopathy) in patients without decompensated cirrhosis at baseline.

	Univariate Analysis		Multivariate Analysis	
Variables	Hazard Ratio (95% CI)	*p* Value	Hazard Ratio (95% CI)	*p* Value
Baseline				
Age (year)	1.005 (0.980–1.031)	0.680		
Sex, male vs. female	1.162 (0.574–2.353)	0.677		
HBeAg, yes vs. no	1.087 (0.560–2.111)	0.805		
NA-naïve, yes vs. no	2.592 (0.802–8.372)	0.111		
TDF vs. entecavir	1.030 (0.534–1.986)	0.930		
Diabetes mellitus, yes vs. no	1.101 (0.544–2.229)	0.789		
Hypertension, yes vs. no	1.284 (0.680–2.422)	0.441		
HBV DNA, per log_10_ IU/mL	0.965 (0.790–1.179)	0.729		
AST, per U/L	1.000 (0.999–1.002)	0.614		
ALT, per U/L	0.998 (0.994–1.001)	0.215		
Total bilirubin, per mg/dL	1.068 (0.844–1.353)	0.582		
Albumin, per g/L	0.262 (0.154–0.448)	<0.001		
INR, per ratio	1.328 (0.412–4.288)	0.635		
Platelet, per 10^3^/μL	0.983 (0.976–0.990)	<0.001		
AFP at baseline, per ng/mL	0.996 (0.985–1.006)	0.412		
M2BPGi, per COI	1.180 (1.082–1.288)	<0.001		
HBcrAg, per log_10_ U/mL	0.987 (0.804–1.211)	0.900		
12 months of treatment				
ALT < 40 U/L, per U/L	0.536 (0.292–0.984)	0.044		
AFP, per ng/mL	1.011 (1.002–1.020)	0.016		
Platelet, per 10^3^/μL	0.980 (0.973–0.987)	<0.001	0.986 (0.979–0.994)	0.001
Albumin, per g/L	0.280 (0.197–0.398)	<0.001	0.439 (0.283–0.679)	<0.001
M2BPGi, per COI	1.349 (1.241–1.466)	<0.001	1.135(1.026–1.256)	0.014
HBcrAg, per log_10_ U/mL	1.078 (0.853–1.363)	0.531		

AFP, alpha-fetoprotein; ALT, alanine aminotransferase; AST, aspartate aminotransferase; CI, confidence interval; COI, cut-off index; HBcrAg, hepatitis B core related antigen; HBeAg, hepatitis B e antigen; HBV, hepatitis B virus; INR, international normalized ratio; M2BPGi, Mac-2 binding protein glycosylation isomer; NA, nucleos(t)ide analogue; TDF, Tenofovir disoproxil fumarate.

**Table 6 cancers-14-05063-t006:** Univariate and multivariate analyses of factors associated with liver related mortality or liver transplantation.

	Univariate Analysis		Multivariate Analysis	
Variables	Hazard Ratio (95% CI)	*p *Value	Hazard Ratio (95% CI)	*p* Value
Baseline				
Age (year)	1.010 (0.988–1.031)	0.377		
Sex, male vs. female	1.096 (0.613–1.960)	0.757		
HBeAg, yes vs. no	1.084 (0.620–1.894)	0.778		
Decompensation, yes vs. no	5.519 (3.352–9.087)	<0.001		
NA-naïve, yes vs. no	1.550 (0.706–3.404)	0.275		
TDF vs. entecavir	0.651 (0.346–1.228)	0.185		
Diabetes mellitus, yes vs. no	1.210 (0.676–2.164)	0.521		
Hypertension, yes vs. no	0.958 (0.542–1.692)	0.882		
HBV DNA, per log_10_ IU/mL	0.948 (0.806–1.117)	0.525		
AST, per U/L	0.999 (0.997–1.001)	0.190		
ALT, per U/L	0.997 (0.994–1.000)	0.041		
Total bilirubin, per mg/dL	1.033 (0.983–1.085)	0.204		
Albumin, per g/L	0.303 (0.219–0.419)	<0.001		
INR, per ratio	1.983 (1.078–3.650)	0.028		
Platelet, per 10^3^/μL	0.984 (0.978–0.989)	<0.001		
AFP at baseline, per ng/mL	0.999 (0.994–1.003)	0.491		
M2BPGi, per COI	1.137 (1.087–1.189)	<0.001		
HBcrAg, per log_10_ U/mL	0.987 (0.821–1.166)	0.806		
12 months of treatment				
ALT < 40 U/L, per U/L	0.763 (0.441–1.321)	0.335		
AFP, per ng/mL	1.011 (1.003–1.019)	0.011		
Platelet, per 10^3^/μL	0.984 (0.978–0.989)	<0.001	0.992 (0.986–0.998)	0.009
Albumin, per g/L	0.280 (0.197–0.398)	<0.001	0.350 (0.250–0.490)	<0.001
M2BPGi, per COI	1.239 (1.175–1.306)	<0.001	1.083 (1.012–1.159)	0.021
HBcrAg, per log_10_ U/mL	1.156 (0.944–1.416)	0.160		

AFP, alpha-fetoprotein; ALT, alanine aminotransferase; AST, aspartate aminotransferase; CI, confidence interval; COI, cut-off index; HBcrAg, hepatitis B core related antigen; HBeAg, hepatitis B e antigen; HBV, hepatitis B virus; INR, international normalized ratio; M2BPGi, Mac-2 binding protein glycosylation isomer; NA, nucleos(t)ide analogue; TDF, Tenofovir disoproxil fumarate.

## Data Availability

All data generated or analyzed in the study are included in the article or its online Appendix A files. Further inquiries can be directed to the corresponding author.

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
