# Peer review of "A Mac-2 Binding Protein Glycosylation Isomer-Based Risk Model Predicts Hepatocellular Carcinoma in HBV-Related Cirrhotic Patients on Antiviral Therapy"

_cancers, 2022, doi:10.3390/cancers14205063_

Round 1
Reviewer 1 Report
Unlike those clinical prediction models constructed based on public databases such as TCGA, this study is highly innovative. Nonetheless, I still have some minor suggestions.
1. The discrimination and calibration of the prediction models need to be described in more detail.
2. More evidence is recommended for the screening of variables based on clinical significance.
3. The article uses too many tables. To increase the readability of the manuscript, I suggest using nomogram to present the final constructed model.
4. The conclusions of the study would have been more credible if external validation had been performed.
Reviewer 2 Report
The focus of the study is to evaluate the predictive performance of ASPAM-B score at HCC. This study based on a cohort collected from 2008 and 2018. The follow up study obtained from the serum HBV DNA and ALT levels. In addition, the lectin immunoassay of Mac-2 binding protein glycosylation isomer (M2BPGi) in serum sample was added into ASPAM-B parameter. Some points need to notice at following:
There are reverse labeled of figure 1 and figure 2, and hard to compare the result description.
Fig 3B is cumulative incidences of liver-related mortality or transplantation. However, in 2.1 patients sections, the transplantation is included in the exclusion criteria. This effect should be clarified.
Table 4 made risk scores ranged from 0~11.5, however no clear description of computation and calculate formulate been summarized in manuscript.
Table 4 performed risk score predict rates were increased significantly when year increased, but not in Table 5. Authors might try to make improvement. For example the total risk score from 6 to 8, the rate increased from 19.5% to 91.7%. However, in development cohort, the value from year 3 to 9 has less difference.
Table 6 and table 7 gave the univariate and multivariate analyses of albumin at 0.1~0.4 hazard ratio at base line level. However, it was not been demonstrated in 12 months of treatment. Due to obvious p-value, here should address the level of albumin in 12 months treatment.
Table 7 mentioned the Cox regression analysis, the lower albumin and higher M2BPGi were found at 12 months. However, the results of AFP tumor biomarker or AST/ALT metabolism makers were not described in detail.
Reviewer 3 Report
It is an interesting manuscript about “A Mac-2 binding protein glycosylation isomer-based risk model predicts hepatocellular carcinoma in HBV-related cirrhotic patients on antiviral therapy”.
My concern is determined in the following points.
(1) M2BPGi levels at 12 months of treatment correlated better with future HCC risk compared to the baseline measurement. 12 months after NA treatment may represent an ideal time point for future refinement of the optimal risk model to predict HCC occurrence in patients with CHB receiving long-term NA treatment.
(2) The M2BPGi-based ASPAM-B risk model exhibited good discriminant function in predicting HCC occurrence and stratified HCC risk in cirrhotic patients who received long-term NA treatment. Risk stratification of HCC occurrence in these patients may assist clinicians with individualization of HCC surveillance program in the antiviral therapy era. In addition to HCC, M2BPGi level at 12 months of treatment was a useful marker to predict cirrhotic events and liver-related mortality in cirrhotic patients receiving NA treatment.
Below mentioned should be referred to;
Serum M2BPGi level at 48 weeks is a useful predictor for HCC development in patients with CHB who receive NA therapy for 48 weeks. Therefore, 12 months of treatment in this study was considered too short period of treatment to evaluate future HCC risk.
In CHB patients receiving long-term antiviral treatment more than 5 years, serum M2BPGi level not only serves as an independent HCC predictor but also complements PAGE-B in stratifying HCC risks.
Serum M2BPGi level significantly decreases after NA treatment in CHB patients. Baseline level can be factored into the risk prediction of HCC in NA-treated patients with cirrhosis.
Baseline and 3-year serum M2BPGi may be useful to identify high risk patients on antiviral treatment for subsequent HCC development.
High serum M2BPGi within 3 years after HBeAg seroconversion was a strong predictor for subsequent HCC development in treatment-naive HBeAg-negative CHB patients.
Therefore, serum M2BPGi levels long term (more than 3 years) after NA therapy should be evaluated to identify high risk patients on antiviral treatment for subsequent HCC development.
